# Inducing Artificial Uncertainty in Language Models

## Abstract

In safety-critical applications, language models should be able to characterize their uncertainty with meaningful probabilities. Many uncertainty quantification approaches require supervised data; however, finding suitable unseen challenging data is increasingly difficult for large language models trained on vast amounts of scraped data. If the model is consistently (and correctly) confident in its predictions, the uncertainty quantification method may consistently overestimate confidence on new and unfamiliar data. Finding data which exhibits enough uncertainty to train supervised uncertainty quantification methods for high-performance models may therefore be challenging, and will increase in difficulty as LLMs saturate datasets. To address this issue, we first introduce the problem of inducing *artificial uncertainty* in language models, then investigate methods of inducing artificial uncertainty on trivially easy data in the absence of challenging data at training time. We use probes trained to recognize artificial uncertainty on the original model, and find that these probes trained on artificial uncertainty outperform probes trained without artificial uncertainty in recognizing real uncertainty, achieving notably higher calibration on hard data with minimal loss of performance on easy data.

## 1 Introduction

Due to increased training-time resources, large language models (LLMs) have increasing capabilities to memorize training data (Leybzon & Kervadec, 2024), and the lack of transparency in the contents of many pretraining datasets (Grattafiori et al., 2024; Achiam et al., 2023; Jiang et al., 2023) means it may be unclear whether an LLM has previously been trained on any particular text before. Already, there are suspicions that some LLMs may have overfit to evaluation data (Wu et al., 2025), and this problem is likely to worsen as LLMs scale. As LLM performance increasingly saturates existing datasets and as it becomes more challenging to find unseen training data, the uncertainty that LLMs tend to have in their predictions decreases. While this is not an issue when the LLM is certain in its predictions and generally correct, it is almost certain that an LLM in real-world use will encounter data that they fail on. In many cases, LLMs' reported uncertainty is less than might be expected from empirical error rate—in other words, they are overconfident in their predictions (Xiong et al., 2024).

One natural solution to this problem is to create more datasets with which LLMs are unfamiliar, and may occasionally fail on. As language models grow more capable, the cost of creating these datasets increases; take, for instance, the Graduate-Level Google-Proof Question Answering dataset (Rein et al., 2023). GPQA was constructed to challenge current state-of-the-art models and required around $120,000[1] to construct its 398 challenging questions. Even benchmarks explicitly designed to resist saturation are not immune: Humanity's Last Exam (Phan et al., 2025), a 2,500-question expert-level benchmark, saw frontier model accuracy rise from below 10% to over 60% within a single year of release. Additionally, datasets constructed to address this problem may later be incorporated into pretraining datasets, necessitating continuing generation of new datasets. For this reason, construction of new datasets is an unsustainable solution to this problem moving forward.

Ideally, training supervised uncertainty quantification methods on a model with worse performance (and thus greater uncertainty in its predictions) and applying them to the original overconfident LLMs would result

---

[1] https://wp.nyu.edu/arg/can-good-benchmarks-contain-mistakes/

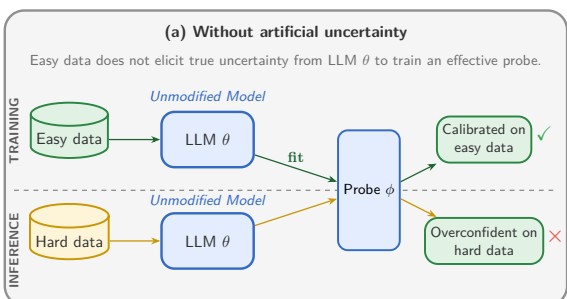 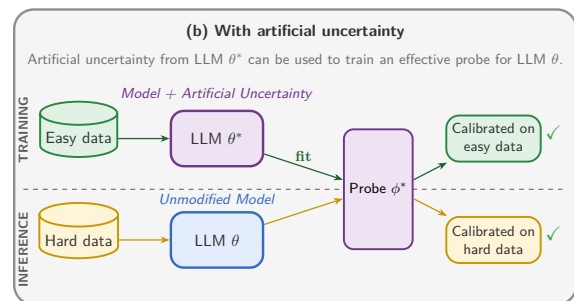

Figure 1: **Overview of our problem.** (a) Due to benchmark saturation or data leakage, the model is confidently correct with little uncertainty. A probe trained to recognize uncertainty on $D_{easy}$ yields an uninformative uncertainty estimates on challenging data, as it does not have a good representation of uncertainty. (b) We construct a model with higher uncertainty, either by using dropout at training time or by using unlearning to remove a model's capabilities. We find that a probe trained on the hidden states from this model on easy data results in calibrated estimates of the **original model's** uncertainty on challenging data, while retaining calibration on easy data similar to that seen at training time.

in more calibrated uncertainty estimates. However, there would almost certainly be significant train-test mismatch between an arbitrary pair of models, as internal representations vary widely based on training and architecture. This motivates constructing a model as close as possible to the one being probed, but with increased uncertainty in its predictions. Here, we examine two straightforward methods of artificially increasing uncertainty in a model's predictions: *dropout* and *unlearning*. Dropout (Srivastava et al., 2014) is often used at training time to prevent overfitting, but may be a convenient method of efficiently reducing model saturation[2]. Unlearning is commonly used for removing known data points from a model's knowledge while impacting overall performance as little as possible (Shaik et al., 2024), which conceptually suggests that it may be possible to create an "unseen" training set from trivially easy or familiar data. We suggest that running a model on an unlearned easy or previously seen dataset simulates the original model encountering a challenging and unseen dataset. As in both of these cases the internal representations are similar to the original model's, we can use the predictions of this unlearned model to learn an uncertainty probe and apply it to the original model.

Our main contributions are as follows:

- We introduce the problem of constructing **artificial uncertainty**, investigating how to induce good representations of uncertainty for language models on data they would otherwise be confident on. To our knowledge, no prior work has formalized this problem or systematically investigated methods of inducing artificial uncertainty for language model calibration.

- We demonstrate that dropout and unlearning cause effective artificial uncertainty representations by training uncertainty probes on models with these interventions and applying them to the original models on challenging test data, resulting in improved calibration (§5). As these probes are applied to the original models, this comes with **no change** to overall accuracy.

- We propose a straightforward unsupervised hyperparameter selection method, which does *not* require challenging validation or calibration data (§6.1).

- We validate our artificial uncertainty methods by doing thorough ablations to determine which experimental choices are important, including comparisons to aleatoric uncertainty (§G), variations in unlearning methods (§A), and layers affected by dropout (§C).

---

[2]Dropout has previously been used in inference-time methods such as MC-Dropout (Gal & Ghahramani, 2016) to increase diversity, indicating it may have the potential to induce uncertainty.

## 2 Preliminaries

### 2.1 Probing

Probing has been used as an uncertainty quantification method, by extracting a models confidence from its internal representations (Azaria & Mitchell, 2023; Slobodkin et al., 2023). Training a simple classifier (generally a single linear layer) to predict the likelihood of correctness from an LLM's hidden states often yields an effective measure of confidence (Kadavath et al., 2022). This provides us with an effective evaluation method for artificial uncertainty, as probes must be trained on supervised data to map an LLM's internal representations to confidence. Without data that elicits uncertainty, probes will be ineffective, and if a language model's representation of artificial uncertainty fails to be similar to actual uncertainty, probes will not generalize.

More formally, given a datapoint $x \in D_{cal}$ and LLM $\theta$, $\theta(x) \to h_\theta, \hat{y}$ where $h_\theta$ is a hidden state (usually the final hidden state) and $\hat{y}$ is the predicted answer. Given the true label $y$, a simple classifier $\phi$ can then be trained using a principled loss function $\mathcal{L}(\phi(h_\theta), \hat{y}, y)^3$, as an objective function to align the probe's predictions to the language model's accuracy.

However, for a probe to learn to recognize that an LLM is uncertain, the training data must elicit uncertainty and some proportion of the labels must be negative. If the training data has been memorized by the LLM, or if the training data is insufficiently challenging and thus saturated, the probe will fail to identify the model's internal representations of uncertainty. In other words, while ideally $D_{cal}$ would have examples with both high and low uncertainty, it may become the case that $D_{cal} \subset D_{easy}$, and all datapoints used to train the probe have high confidence.

### 2.2 Aleatoric vs. epistemic uncertainty

Uncertainty can be broadly decomposed into two types: *aleatoric* uncertainty (data uncertainty), and *epistemic* uncertainty (model uncertainty) (Kiureghian & Ditlevsen, 2009). Larger and more powerful models tend to have lower epistemic uncertainty; in many cases, large models can improve on examples where small models exhibit uncertainty (Narayan et al., 2022). However, aleatoric uncertainty is irreducible and cannot be improved by increasing model size or training. A potential method of introducing artificial uncertainty is to intentionally provide the LLMs with ambiguous or unanswerable questions to induce uncertainty. This ensures that the LLM will always be unable to answer questions. However, this induces *aleatoric* uncertainty, rather than epistemic uncertainty. We investigate whether this distinction matters for artificial uncertainty in Appendix G.

## 3 Methods

### 3.1 Problem setup

Supervised uncertainty quantification methods are generally trained using an unseen calibration or training set $D_{cal}$, which can be used to learn to determine a model's confidence or uncertainty. This relies on the assumption that this data is both *unseen* and *challenging*, and thus not always correctly confident. However, as the scale of pretraining data and the capabilities of models improve, this assumption increasingly fails to hold true. We propose that this problem may be resolved through methods of inducing *artificial* uncertainty, which resolve the need for *true* uncertainty caused by unfamiliar and challenging data. In order for artificial uncertainty to be useful, it must demonstrate similarity to true uncertainty. It should be possible for uncertainty quantification method trained to recognize and quantify artificial uncertainty to be applied to the *original* model, to avoid decreasing accuracy or effectiveness on the original task. We consider probing for uncertainty quantification as a method of evaluating the similarity of artificial uncertainty to true uncertainty: if probes trained to recognize artificial uncertainty from a modified model's internal representations recognize true uncertainty in the original model, this demonstrates similarity between artificial and true uncertainty.

---

[3]In the main experiments, we use binary cross-entropy.

As frontier models continue to improve, saturation of evaluation datasets will become an increasingly pressing issue. To simulate this future situation using open-weight language models in a controlled setting, we make the following experimental assumptions:

**Assumption 1.** *We have a large amount of easy data, $D_{easy}$, where $\theta$ will consistently answer correctly and with high confidence. We do not have hard data: questions that LLM would get wrong but where the true answer is known.*

**Assumption 2.** *At test time, the LLM may encounter some unknown amount of challenging examples $D_{hard}$.*

We emphasize that this is a **simulation** of this outcome, and that we aim for this simulation to act as a proof of concept motivating future work for frontier models. The goal of these assumptions is to provide a framework where we can test methods of generating artificial uncertainty on small-scale models realistically.

### 3.2   Probing for uncertainty

We consider two variants of LLM probes. We use P(IK) from Kadavath et al. (2022), a linear layer predicting the model's confidence from the hidden state after the last token. Therefore, this creates a train/test mismatch, as the learned classifier is trained on final hidden states from the modified model but applied to final hidden states from the original base model. We also use a variant based on Kossen et al. (2024) where we learn this linear layer using the hidden state immediately after the question, predicting the likelihood that the model can accurately answer the question before generating the answer. Formally, at training time, we construct a single linear layer $\phi$ as a probe and use a language model $\theta'$ which has been altered in some way to create artificial uncertainty. We then train $p$ on the accuracy label of easy data to minimize the binary cross-entropy (BCE) loss, thus training P(IK) to accurately predict the likelihood of a correct answer on the modified model, given an easy dataset $D_{easy}$ of $N$ $(x, y)$ pairs where $x$ is the question and $y$ is the correctness of the corresponding answer:

$$\mathcal{L}(\phi) = \frac{1}{N} \sum_{i=1}^{N} \text{BCE}\Big(\phi(h_{\theta'}(x_i)), y_i\Big), (x, y) \in D_{easy}$$

At inference time, we then apply the trained model $\phi$ to the hidden states of the **original** language model $\theta$ rather than the modified language model $\theta'$, and apply it to hard data $D_{hard}$ rather than the trivially easy dataset:

$$P(\text{IK}|x) = \phi(h_{\theta}(x)), x \in D_{hard}$$

### 3.3   Artificial uncertainty methods

We investigate two methods of introducing model uncertainty to the correctly confident language model $\theta$ in order to learn a more effective probe.

**Unlearning**   With unlearning, our goal is to selectively increase uncertainty on $D_{easy}$ to train a calibration probe that will transfer to $D_{hard}$. Unlearning should result in a model $\theta'$ with higher epistemic uncertainty.

Although probes are trained on predictions from $\theta'$, they are applied to the base model $\theta$ and requires $\theta'$ remaining close to $\theta$. If unlearning causes large deviations, then a calibration map learned from $\theta'$ may not generalize to $\theta$. Because of this, we select unlearning procedures that introduce small localized perturbations to the model, increasing uncertainty on selected inputs while preserving its overall predictive behavior, rather than attempting to fully erase knowledge or approximate the distribution of a retrained model.

We use data from $D_{easy}$ to construct an unlearning set $U$. This probe is then evaluated on predictions from the original model $\theta$ and applied to examples from $D_{hard}$. This separation ensures that calibration performance reflects generalization induced by unlearning, rather than direct exposure to the calibration data.

In principle, many existing unlearning methods could be applied in this setting. However, our empirical results show that methods designed to tightly constrain deviation from a reference model or explicitly bound degradation tend to induce only modest changes in predictive uncertainty. In contrast, by directly increasing the training loss on selected examples via gradient ascent, we induce larger and more reliable increases in uncertainty while remaining stable through monitoring and early stopping. As a result, we adopt gradient ascent as our primary unlearning mechanism and defer discussion of alternative methods to Appendix A. [4]

We perform gradient ascent directly on the loss of the forget set $U$ while applying $\ell_2$ regularization to help constrain parameter drift:

$$\theta_{t+1} = \theta_t + \lambda_U \nabla_{\theta_t} \ell(h_{\theta_t}(x_U), y_U)$$

Additionally, we optionally restrict the unlearning loss to the final K tokens of the assistant completion, providing a simple mechanism for localizing the unlearning signal to the model's response without relying on dataset-specific answer annotations. While this may affect both explanatory and answer tokens depending on output structure, the resulting increase in training loss leads to higher predictive uncertainty on held-out examples, which we find sufficient for inducing controlled increases in predictive uncertainty.

Unlike unlearning methods that rely on a frozen reference model or explicit distributional constraints, gradient ascent directly modifies the model's loss landscape on selected inputs, enabling stronger and more targeted changes in predictive uncertainty. Although this approach is more aggressive, we find that during experimental runtime, using conservative unlearning rates and monitoring training/early stopping allow us to avoid gradient explosion and preserve overall predictive performance.

**Dropout** Another method of constructing $\theta'$ is to cause worse performance by adding dropout (Srivastava et al., 2014) to the attention layers[5]. This effectively adds random noise, degrading accuracy for the predictions used to train the probe. This has the advantage of improved efficiency at training time, and reduced hyperparameter search required, as only the amount of dropout needs to be tuned.

### 3.4 Unsupervised hyperparameter selection

Per Assumption 1, we do not have validation data suitable to fit hyper-parameters. Therefore, we require an alternate method of *unsupervised* hyperparameter selection to pick the dropout rate and unlearning extent. While challenging data may be difficult to acquire, we can readily evaluate models on easy data. Note that one reason behind decreased performance on hard data is a lack of *diversity* in predictions. Therefore, we propose using the variance of the probe's predictions on the calibration set as a hyperparameter selection technique. That is, given a set hyperparameters $H$, we select the $h \in H$ that maximizes the variance in the predictions $\hat{p}^{(h)}$ from probes trained on the calibration set:

$$h^* = \underset{h \in H}{\operatorname{argmax}} \left\{ \mathbb{V}[\hat{p}^{(h)}] \right\}$$

Intuitively, compared to a baseline which is always confidently correct ($\mathbb{V}[\hat{p}^{(h)}] \approx 0$), we are searching for $h^*$ that induce a *spectrum* of confidences in the probes' predictions on the calibration set. This enables the probe to discriminate between hidden states that are associated with more or less uncertainty. We find that the variance of a trained probe's predictions on easy data correlates well with performance, as we discuss further in §6.1.

---

[4]We apply this procedure with full fine-tuning for `Llama-3.1-8B-Instruct` and using LoRA (Hu et al., 2021) for `gemma-3-12b-it` and `phi-4`.

[5]In Appendix C, we demonstrate that adding dropout to the attention layers rather than later layers significantly improves the effects of artificial uncertainty.

## 4 Experiments

We lay out a series of experiments to answer two main questions: how can we induce sufficient artificial uncertainty while maintaining similarity to the original model, and how can we select hyperparameters without supervised challenging data.

### 4.1 Datasets

We examine the effects of adding artificial uncertainty in a multiple-choice question answering setting, specifically in the STEM domains.

**Calibration set($D_{easy}$)** In the STEM area, we use examples from the SciQ dataset (Welbl et al., 2017). LLMs achieve high performance on SciQ (Sanchez et al., 2023), and the dataset predates many existing models, making prior exposure through pretraining data plausible. We use the existing splits, but further split the validation set into 500 validation examples and 500 calibration examples. During unlearning, we consider all 11,679 examples from the training set as our unlearning set.

To allow the unlearned model to see examples with low uncertainty, $D_{easy}$ also includes 500 examples from the training set of CommonsenseQA (Talmor et al., 2019), an easy multiple choice question answering dataset which revolves around answering simple common-sense questions. As we have not specifically unlearned these examples, LLM performance should remain high.

To ensure that even our least powerful model, `Llama-3.1-8B-Instruct`, can achieve saturated performance, we filter $D_{easy}$ to only questions that Llama-3.2-3B-Instruct answered correctly with a different random seed from the one used in our experiments, giving us a final dataset of 1000 questions[6] that any of our LLMs should answer with very high accuracy and confidence. Our models all achieve over 90% accuracy on this set, producing a rough approximation of our setting. Although we note that a simulation of a saturated model would achieve perfect accuracy on our training set, outperforming a baseline probe which has only been on trained on a single class is trivially easy and thus a weak baseline. We therefore consider this a weakened version of Assumption 1, where while data may not be hard, it may still be answered incorrectly[7].

**Test sets ($D_{hard-MMLU}$, $D_{hard-GPQA}$, $D_{easy-ARC}$)** $D_{hard}$ should have some amount of questions that are difficult for an LLM to answer. To accomplish this, we choose two challenging datasets where our LLMs are expected to have high model uncertainty. The first is GPQA (Rein et al., 2023), a multiple-choice QA dataset written by domain experts. This dataset presents a significant challenge: more powerful models than those tested here achieve less than 40% accuracy. We reserve the diamond set (selected for annotator agreement) as our test data, and all other datapoints may be used for validation data; for evaluation, we use the 198 examples in the diamond set.

The second challenging dataset is MMLU-pro (Wang et al., 2024b), a more challenging version of MMLU (Hendrycks et al., 2020) which has increased distractor options and requires more reasoning. We use previously constructed splits[8]. We narrow the data to STEM topics[9].

In §6.2, we use an easy test set, ARC-Easy (Clark et al., 2018). This is a dataset of grade-school multiple choice science questions, which should be easy for our comparatively small LLMs to answer. We use the existing test split of ARC-Easy.

**Validation set** In our problem, we assume there is no available challenging data available to fit hyper-parameters. We discuss how to select hyperparameters without validation data in §6.1; however, to show correlation of performance to this procedure, we require validation sets. We emphasize that we select our final models *without the use of this validation data*, as this knowledge would not be available in our actual setting. For one validation set, we use examples from GPQA. We select 100 examples randomly from the

---

[6]We will release these questions along with our code at publication.
[7]We demonstrate effects when this assumption holds completely true with 100% accurate data in Appendix I
[8]https://huggingface.co/datasets/answerdotai/MMLU-SemiPro
[9]Math, health, chemistry, physics, engineering, biology, computer science

non-diamond set as our validation set. We use an additional validation set of 1000 STEM examples from the validation split[10] of MMLU-pro.

## 4.2 Baseline

Our baseline is a probe trained on the original model *without artificial uncertainty*. As described in §4.1, we strengthen this baseline by retaining some errors in the probe training data rather than filtering to perfect accuracy. We are not aware of prior work that addresses probe training under Assumption 1, where no challenging data is available at any stage of the pipeline. Post-hoc calibration methods such as temperature scaling (Guo et al., 2017), Thermometer (Shen et al., 2024), and adaptive temperature scaling (Xie et al., 2024) rescale the model's own output probabilities rather than training a separate uncertainty predictor, and assume that the calibration data contains variation in model confidence, which is absent under Assumption 1. Our setting therefore differs from previous approaches to improving calibration: the challenge is *creating* uncertainty signal where none naturally exists rather than calibrating existing uncertainty estimates.

## 4.3 Models and metrics

**Models** For our language models, we use `Llama-3.1-8B-Instruct` (Grattafiori et al., 2024), `Phi-4` (14B parameters) (Abdin et al., 2024), and `gemma-3-12B-it` (Team et al., 2025).

**Metrics** There are two main factors evaluated in uncertainty estimates: calibration, or the correspondence of predicted confidence to error rate, and discrimination, which represents the model's ability to distinguish between positive and negative classes. We consider Brier score (Brier, 1950) as our main metric, which combines aspects of both discrimination and calibration to give a holistic evaluation of the effectiveness of the uncertainty estimate. We also report calibration using Expected Calibration Error (Guo et al., 2017), a commonly used error metric where successful models have low ECE, and discrimination using Area Under the Reciever Operating Curve (AUROC), where a perfect ranking achieves AUROC of 1 and random chance achieves AUROC of 0.5. We also report accuracy on each of our test sets to give context for the level of divergence from the training setting. We calculate exact-match accuracy using string operations (isolating the answer using a standard tag format, removing punctuation, standardizing case)[11].

## 5 Results

Table 1 shows the results for all configurations of our experiments. We first note that probe performance degrades noticeably as accuracy decreases: this is clear when examining `phi-4` on MMLU-pro, which achieves accuracy of 0.680. The base probe achieves 0.219 or 0.265 Brier score, depending on whether the probe is trained to predict confidence before or after the answer. However, the very same base probes achieve 0.312 and 0.396 Brier score when tested on GPQA, which only achieves accuracy of 0.551. This validates our experimental assumption: that probes trained on easy data fail to generalize to hard data.

We find that introducing artificial uncertainty is generally successful in improving probe performance. In particular, we find that dropout most reliably improves the Brier score of probes, outperforming the base probes in 11 out of 12 settings and outperforming all methods in 9. Unlearning is also generally successful in improving probe performance, outperforming the base probe in 10 out of 12 settings and outperforming all methods in 3 settings. However, dropout appears to induce more substantial drops in Brier score in most settings; considering this as well as the context that unlearning requires significantly more computational expense to perform, we suggest that dropout is the more practical choice to artificially introduce uncertainty for challenging data[12].

Analyzing calibration (ECE) vs discrimination (AUROC), we find that the improvements in Brier score generally come from improvements in calibration, with substantial drops in ECE. In 6 out of 12 cases, we find that AUROC drops for probes trained with dropout, with the remaining cases improving AUROC.

---

[10]We use the split described in `https://huggingface.co/datasets/answerdotai/MMLU-SemiPro`
[11]We present our prompting scheme in Appendix J
[12]Though we find that unlearning may be more robust to easy data as seen in §6.2.

| Dataset | Model | Accuracy | Position | Method | Brier↓ | AUROC ↑ | ECE ↓ |
|---------|-------|----------|----------|--------|--------|---------|-------|
| GPQA | Llama-8B | 0.247 | Before | Base | 0.719 | 0.550 | 0.728 |
| | | | | Unlearned | 0.490 | 0.597 | 0.546 |
| | | | | Dropout | **0.267** | 0.592 | 0.275 |
| | | | After | Base | 0.494 | 0.685 | 0.551 |
| | | | | Unlearned | 0.320 | 0.532 | 0.306 |
| | | | | Dropout | **0.227** | 0.624 | 0.179 |
| | Phi-4 | 0.551 | Before | Base | 0.312 | 0.502 | 0.250 |
| | | | | Unlearned | 0.314 | 0.482 | 0.236 |
| | | | | Dropout | **0.300** | 0.571 | 0.231 |
| | | | After | Base | 0.396 | 0.591 | 0.399 |
| | | | | Unlearned | 0.323 | 0.582 | 0.294 |
| | | | | Dropout | **0.292** | 0.574 | 0.237 |
| | Gemma-12B | 0.323 | Before | Base | 0.663 | 0.466 | 0.667 |
| | | | | Unlearned | 0.605 | 0.493 | 0.620 |
| | | | | Dropout | **0.585** | 0.487 | 0.585 |
| | | | After | Base | 0.598 | 0.486 | 0.610 |
| | | | | Unlearned | **0.330** | 0.417 | 0.324 |
| | | | | Dropout | 0.531 | 0.529 | 0.533 |
| MMLU-pro | Llama-8B | 0.366 | Before | Base | 0.552 | 0.558 | 0.565 |
| | | | | Unlearned | 0.376 | 0.534 | 0.347 |
| | | | | Dropout | **0.252** | 0.563 | 0.125 |
| | | | After | Base | 0.423 | 0.741 | 0.455 |
| | | | | Unlearned | 0.281 | 0.631 | 0.212 |
| | | | | Dropout | **0.219** | 0.708 | 0.100 |
| | Phi-4 | 0.680 | Before | Base | **0.219** | 0.636 | 0.106 |
| | | | | Unlearned | 0.225 | 0.621 | 0.120 |
| | | | | Dropout | 0.241 | 0.542 | 0.141 |
| | | | After | Base | 0.265 | 0.683 | 0.261 |
| | | | | Unlearned | **0.206** | 0.719 | 0.144 |
| | | | | Dropout | **0.206** | 0.655 | 0.086 |
| | Gemma-12B | 0.526 | Before | Base | 0.453 | 0.467 | 0.448 |
| | | | | Unlearned | 0.409 | 0.554 | 0.399 |
| | | | | Dropout | **0.365** | 0.492 | 0.324 |
| | | | After | Base | 0.387 | 0.732 | 0.406 |
| | | | | Unlearned | **0.269** | 0.676 | 0.176 |
| | | | | Dropout | 0.301 | 0.694 | 0.274 |

Table 1: Comparing the base probes to probes trained using artificial uncertainty. Introducing artificial uncertainty dramatically improves performance on our challenging test sets, largely through large improvements in calibration. "Before" and "After" refer to whether the probe was trained on a pre-generation state or a post-generation state. We **bold** the best-performing Brier score.

Unlearning improves AUROC in only 4 out of 12 cases. This finding indicates that artificial uncertainty is unlikely to significantly improve a model's discriminative capabilities on challenging data. In this sense, artificial uncertainty methods may be considered similar to post-hoc calibration, but *without the need for calibration data*, indicating that these methods may be used without unseen calibration data.

# 6 Analysis

## 6.1 Unsupervised hyperparameter selection

As part of our first research question, we investigate how much artificial uncertainty is necessary to replicate true uncertainty—specifically, amount of dropout at training time, or steps of unlearning. We compare standard deviation on 500 easy examples from SciQ (Welbl et al., 2017) against Brier score on our GPQA validation set. We show an example of the results for Llama-8B in Figure 2, demonstrating how diversity of predictions on easy data may be used to predict model performance. For our main-text experiments, we selected the classifiers within the range of 0.02-0.20 which achieved the highest standard deviation on this easy validation set, rather than using the validation set; this demonstrates that without a challenging validation set, it is still possible to select models with artificial uncertainty.

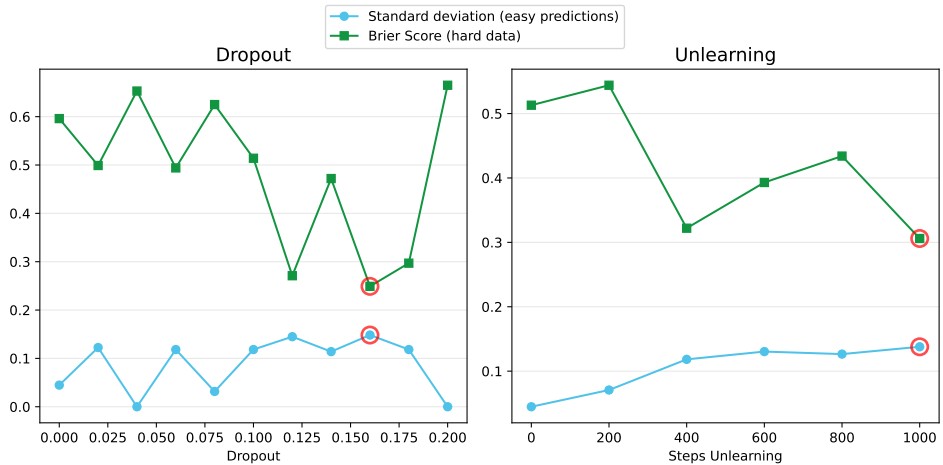

Figure 2: Comparing Brier score performance on validation data (which would not be easily available) to the standard deviation of predictions on easy test data (which is easy to obtain) for Llama 8B Instruct. Notably, there is a clear relation between high standard deviation and lower Brier score. We highlight the points with the highest variance on the validation data and their corresponding Brier score; this is the selection criteria for extent of artificial uncertainty we use.

## 6.2 Evaluation on non-challenging data

Our main results focus on performance in extreme environments designed to challenge a language model. Here, we discuss the impact of our artificial uncertainty methods when there is little to no domain shift. This is important to consider, as large SOTA models which perform well on benchmarks are more likely to encounter queries they can answer than queries they cannot. Table 2 shows the results of our artificial uncertainty-trained probes with our easy test dataset, ARC-easy (Clark et al., 2018). In this environment, while there is a mild domain shift (slightly different datasets, slightly different accuracies), at test time the accuracies should remain close to the training-time dataset. An intuitive result would indicate that better calibration on hard data would result in correspondingly *worse* calibration on easy data. However, we here find that this does not always hold true; in fact, we find that in every setting, a probe trained with artificial uncertainty outperforms or matches the base model, potentially indicating that the introduced diversity improves the model's ability to differentiate between answers even at high confidence levels. We also note that unlearning appears to outperform dropout in most cases, suggesting that the small degree of divergence from the original model may allow it to perform well on easy data. This may be a reason to use unlearning despite the relatively high computational cost and comparatively worse performance to dropout on challenging data, as improved robustness on easy data is likely to be valuable in most settings.

| Dataset | Model | Accuracy | Position | Method | Brier↓ | AUROC ↑ | ECE ↓ |
|---------|-------|----------|----------|--------|--------|---------|-------|
| ARC | Llama-8B | 0.929 | Before | Base | 0.070 | 0.596 | 0.066 |
| | | | | Unlearned | **0.066** | 0.635 | 0.041 |
| | | | | Dropout | 0.082 | 0.581 | 0.107 |
| | | | After | Base | 0.064 | 0.824 | 0.054 |
| | | | | Unlearned | **0.063** | 0.818 | 0.066 |
| | | | | Dropout | 0.099 | 0.791 | 0.199 |
| | Phi-4 | 0.979 | Before | Base | 0.026 | 0.739 | 0.065 |
| | | | | Unlearned | **0.020** | 0.719 | 0.014 |
| | | | | Dropout | 0.119 | 0.695 | 0.267 |
| | | | After | Base | **0.020** | 0.838 | 0.008 |
| | | | | Unlearned | **0.020** | 0.719 | 0.014 |
| | | | | Dropout | 0.054 | 0.878 | 0.181 |
| | Gemma-12B | 0.971 | Before | Base | **0.028** | 0.673 | 0.019 |
| | | | | Unlearned | **0.028** | 0.693 | 0.020 |
| | | | | Dropout | 0.052 | 0.565 | 0.117 |
| | | | After | Base | 0.028 | 0.771 | 0.027 |
| | | | | Unlearned | 0.059 | 0.780 | 0.099 |
| | | | | Dropout | **0.026** | 0.762 | 0.014 |

Table 2: Performance on a comparably easy dataset to SciQ, the training data with our methods. We find that unlearning, in many cases, outperforms the base model even on this, indicating this may be a more reliable method of calibrating probes than dropout if the model is likely to encounter few examples of challenging data.

Examining these results along with the reliability diagrams found in Figure 4 and discussed in Appendix F, we note that the decrease in calibration on easy data tends to be much less notable than the improvement in calibration on hard data.

## 7 Related work

**Uncertainty quantification**   Natural language presents a challenge in uncertainty quantification for several reasons, such as semantic equivalence of different generations, compute required for sampling approaches, and input ambiguity (Liu et al., 2025). A variety of approaches have been considered to quantify model confidences, such as using information from token-level probabilities (Lakshminarayanan et al., 2017; Kadavath et al., 2022), measuring sample variance (Xiong et al., 2024; Kuhn et al., 2023), prompting (Tian et al., 2023; Xiong et al., 2024), or fine-tuning a model to verbalize its confidence (Chaudhry et al., 2024; Hager et al., 2025). While dropout has previously been used in the context of uncertainty, for instance in MC dropout(Gal & Ghahramani, 2016), it has generally been used at inference time to diversify predictions. Our method, instead, applies dropout at training time, allowing for the use of dropout to increase uncertainty while both using a single sample and additionally not impacting accuracy of the model at inference time.

**Probing**   Researchers have trained probes of hidden states for a variety of tasks, including detection of untruthfulness (Marks & Tegmark, 2024), toxicity (Wehner & Fritz, 2025), and knowledge conflict (Gao et al., 2025). The success of probing demonstrates that a model's hidden states contain robust representations of many useful concepts. This also holds true for uncertainty, as first demonstrated by Kadavath et al. (2022) with P(IK), a linear probe trained to determine a model's confidence by quantifying the probability that the model knows the answer to the question. Kossen et al. (2024) extend this concept to work for unsupervised data, by calculating the semantic entropy (Kuhn et al., 2023) and training a linear probe to predict this uncertainty, effectively creating supervised data from unannotated text. They also demonstrate that probes can effectively get signal from the model's hidden state immediately after the question, showing that the model has a concept of whether it can answer the question even before generating its answer. Slobodkin

et al. (2023) demonstrate success when training probes to differentiate between answerable and unanswerable questions, which effectively quantifies *aleatoric* uncertainty.

**Calibration**  Ideally, an uncertainty estimate is well-calibrated, meaning that the predicted uncertainty correlates well with the likelihood a model is correct (Guo et al., 2017). Calibration methods can be used to improve the reliability of uncertainty quantification (Wang, 2024). Many of these methods require suitable calibration data. For instance, Wang et al. (2024a) consider the calibration of verbalized expressions of certainty such as "maybe" by treating each such expression as a distribution over the probability simplex. In this framework, optimal transport can be used to post-hoc calibrate these expressions, which requires calibration data. Manggala et al. argue that traditional notions of calibration are inadequate for question answering, and instead suggests a groupwise post-hoc calibration procedure that conditions on groups of question-answer pairs.

Recent work has sought to reduce the dependence on task-specific calibration data. Shen et al. (2024) train an auxiliary model on data from multiple tasks to predict temperature scaling parameters for new tasks, enabling cross-task generalization without per-task calibration sets. Xie et al. (2024) propose adaptive temperature scaling, which predicts a per-token temperature from the model's hidden features to address calibration degradation after RLHF. However, both approaches fundamentally rescale the model's existing output probabilities and assume that the calibration data contains meaningful variation in model confidence; they do not address the setting where the model is consistently confident across all available data.

Beyond natural language, image mixup has been used to create synthetic images for purposes of improving the calibration of image classifiers (Bouniot et al., 2023). Similarly, Chandu et al. (2024) use image in-filling to introduce epistemic uncertainty into a visual question-answering dataset, finding that visual language models are unable to characterize their uncertainties. These vision-domain approaches are the closest in spirit to our work, as they construct synthetic data to expose calibration failures, but they do not address the transfer of artificial uncertainty from a modified model to the original.

**Unlearning**  Most commonly, unlearning is used in the context of data privacy (e.g. removing identifiable personal data from a model) (Zhao et al., 2024) or removing potentially harmful capabilities (e.g. removing the model's knowledge of bioweapons or cyberattacks) (Li et al., 2024). In these settings, the central objective is for the model to behave as if the target data (forget set) were never observed. This is commonly formalized as approximating the distribution of a model retrained from scratch on the remaining data, excluding the forget set. As a result, unlearning techniques are typically evaluated based on their ability to (i) drive forget-set performance towards random chance or toward the performance of a model trained without exposure to the forget set data, while (ii) preserving performance on a retained dataset (Zhao et al., 2024). For example, methods such as Negative Preference Optimization (NPO) define forgetting relative to a frozen reference model by penalizing increases in sequence log-likelihood on forget examples while using a bounded objective to limit the magnitude of degradation that can be induced (Zhang et al., 2024).

To our knowledge, unlearning has not been explicitly studied with artificial uncertainty as a primary objective. Instead, prior work primarily evaluates success in terms of recoverability. In the context of data privacy and harmful capabilities, it is important that the model *completely forgets* task-specific knowledge, such that relearning time, or even the possibility of recovery, can be used to assess whether unlearning has been successful (Nguyen et al., 2024). This task has been shown to be difficult while maintaining overall model performance. In contrast, our work reframes unlearning as a mechanism for shaping model uncertainty rather than eliminating data recoverability or approximating a retrained distribution. Instead, we deliberately induce forgetting that increases uncertainty on the target inputs. From this perspective, unlearning acts more as a controlled perturbation to the model's confidence landscape rather than an irreversible deletion. This reframing is motivated by the observation that for uncertainty estimation, the objective fundamentally differs from that of traditional unlearning applications. The goal is not complete/catastrophic forgetting, but to create calibrated distinctions between outputs the model should answer confidently and those for which it would express uncertainty.

# 8  Conclusion

In this paper, we have presented a solution for a rapidly approaching problem where lack of unseen and challenging data cause current uncertainty methods to fail to work effectively, as they will not be able to identify the underlying representations of uncertainty without examples of an uncertain model. Our exploration of this problem indicates that it is possible to improve the calibration of the uncertainty estimates by artificially inducing uncertainty, with minimal loss of discriminative ability. In particular, dropout effectively increases calibration dramatically on hard data, at the expense of generally slightly worse performance on easy data. Unlearning generally provides modest improvements to calibration of both easy and challenging data, suggesting that this method may be more effective when the model is unlikely to encounter challenging data. In both cases, probes trained to recognize artificial uncertainty achieved higher calibration on real uncertainty, suggesting that it has similar characteristics to real uncertainty.

Our analyses demonstrate that this is a non-trivial result which is applicable across a variety of models, datasets, and settings. Our results from Appendix C, Appendix G, and Appendix A show that artificial uncertainty cannot simply be constructed through any method that decreases accuracy. Methods which influence later layers of the model tend to be outperformed by methods influencing earlier layers of the model. Similarly, train-test mismatches like attempting to recognize epistemic uncertainty with a probe trained to recognize aleatoric uncertainty are ineffective in improving performance. We also note that, per our scaling analysis in Appendix H, when corrected for accuracy, larger language models tend to have worse calibration than smaller models, indicating that this problem may persist and worsen as models increase in scale. However, probes trained with artificial uncertainty continue to recognize true uncertainty in these large models, indicating that our approach is less affected by scaling the number of parameters. We believe this demonstrates that artificial uncertainty will have increased utility in the future in calibrating supervised uncertainty quantification methods.

A major contribution of this paper is the concept of introducing artificial uncertainty. To our knowledge, no prior work has identified this as a research priority. However, with natural model uncertainty becoming increasingly challenging to elicit in frontier models, as seen by the challenges in creating difficult benchmarks such as GPQA (Rein et al., 2023), we present this work as a first step in attempting to elicit model uncertainty without requiring challenging data at any step in the pipeline. There are many directions for future research; while we focus on two promising methods, we hope that this work will inspire further work to investigate other methods of inducing artificial uncertainty, which may have their own advantages and trade-offs.

**Broader Impact Statement**

The goal of introducing artificial uncertainty is to increase reliability of uncertainty estimates. However, we note that one potential risk is that increased user trust in systems with improved confidence estimates may create misplaced trust or overreliance when the model is still incorrect We still believe that, as a whole, improving uncertainty estimates is beneficial to AI safety.

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

## A  Other unlearning methods

We perform initial experiments with several alternate methods of unlearning.

### A.1  NPO

Unlike direct gradient ascent, *Negative Preference Optimization* (NPO) defines forgetting relative to a frozen reference model. Rather than directly increasing loss on the forget set, NPO penalizes the current model when it assigns higher sequence probability to forget examples than the reference model does, while preserving general capabilities through separate retain updates.

For a forget example $(x_U, y_U) \sim U$, let

$$\log p_{\theta_t}(y_U \mid x_U) \quad \text{and} \quad \log p_{\theta_0}(y_U \mid x_U) \tag{1}$$

denote the sequence log-probabilities assigned by the current model and the frozen reference model, respectively. We define their difference as

$$\Delta(x_U, y_U) = \log p_{\theta_t}(y_U \mid x_U) - \log p_{\theta_0}(y_U \mid x_U), \tag{2}$$

and optimize the forget objective

$$\mathcal{L}_{\text{forget}} = \frac{2}{\beta} \operatorname{softplus}(\beta \, \Delta(x_U, y_U)), \tag{3}$$

where $\beta$ is a tunable coefficient controlling the sharpness of the forgetting penalty. The current model is then updated by gradient descent on this objective:

$$\theta_{t+1/2} = \theta_t - \eta_U \nabla_{\theta_t} \mathcal{L}_{\text{forget}}. \tag{4}$$

After each forget update, we perform $k$ retain updates on retain-set examples $(x_R, y_R) \sim R$ using the standard causal language modeling loss

$$\mathcal{L}_{\text{retain}} = \ell(h_\theta(x_R), y_R), \tag{5}$$

followed by

$$\theta_{t+1} = \theta_{t+1/2} - \eta_R \nabla_{\theta_t} \mathcal{L}_{\text{retain}}. \tag{6}$$

Intuitively, the forget objective penalizes the current model when it assigns higher sequence likelihood to forget examples than the frozen reference model, thereby encouraging the model to move away from the reference model's behavior on the forget set. The retain steps then restore or preserve performance on unrelated retained knowledge. In our implementation, we alternate one NPO forget step with $k$ retain steps, and tune the forget learning rate $\eta_U$, retain learning rate $\eta_R$, retain-step ratio $k$, and NPO coefficient $\beta$. We additionally use gradient accumulation and gradient clipping for stability.

### A.2 RMU

While NPO operates in output space, *Representation Misdirection Unlearning* (RMU) performs forgetting by directly modifying the model's internal hidden representations. Rather than encouraging divergence in token-level predictive distributions or suppressing the likelihood of forget examples, RMU steers the hidden representation of forget-set examples toward a random target direction while preserving retain-set representations relative to a frozen reference model (Li et al. (2024)).

$$\mathcal{L}_{\mathrm{RMU}} = \left\| M_{\theta_t}^{(\ell)}(x_U) - c \cdot \mathbf{u} \right\|^2 + \beta \cdot \left\| M_{\theta_t}^{(\ell)}(x_R) - M_{\theta_0}^{(\ell)}(x_R) \right\|^2, \tag{7}$$

where $M_{\theta}^{(\ell)}(x)$ denotes the hidden activation at layer $\ell$ under model parameters $\theta$, $\mathbf{u}$ is a randomly sampled unit vector, $c$ is a steering coefficient controlling the target norm of the forget representation, and $\beta$ weights the retain objective. Parameters are updated by gradient descent on $\mathcal{L}_{\mathrm{RMU}}$.

The steering loss is computed at a target layer $\ell$, which is treated as a tunable hyperparameter. Gradients from this objective are propagated backward to a specified set of trainable layers, allowing earlier layers to influence the representation at layer $\ell$ while keeping the representational target localized. In our implementation, the target layer, updated layers, parameter subsets, and steering coefficient are tuned independently, enabling control over how localized or distributed the unlearning updates are.

## B Comparison of Alternative Unlearning Methods

This section reports additional results for small-scale experiments on `Llama-3.2-3B-Instruct`, using the alternate unlearning methods that we describe above and evaluating with a small mixed-difficulty validation dataset consisting of the 200 examples from our GPQA validation set described in §4.1 and 200 examples from the validation split of ARC (Clark et al., 2018). Our objective was to find the approach most suitable for inducing artificial uncertainty, especially when considering larger models ($> 30B$ parameters). In particular, we evaluated NPO and RMU against direct gradient ascent, and also compared full-weight unlearning against LoRA-based unlearning to assess whether parameter-efficient updates were sufficient for this objective. We considered both calibration quality and preservation of overall task performance. As shown in Table 3, full-weight gradient ascent provided the most favorable tradeoff, followed by LoRA gradient ascent, motivating their use in our main experiments.

| Method | Variant | Cal. AUROC ↑ | Bin. Cal. Error ↓ |
|---|---|---|---|
| BASE | PRETRAINED | 0.687 | 0.230 |
| GRADIENT ASCENT | FULL, $3 \times 10^{-7}$ | **0.748** | **0.077** |
| GRADIENT ASCENT | LoRA, $1 \times 10^{-5}$ | 0.741 | 0.169 |
| NPO | $4 \times 10^{-6}$, $\beta = 0.1$ | 0.611 | 0.266 |
| RMU | $3 \times 10^{-5}$ | 0.679 | 0.172 |

Table 3: Comparison of candidate unlearning methods on Llama-3B for selecting the primary artificial uncertainty method. Full-weight gradient ascent produced the strongest improvement in calibration error while also improving calibration AUROC, outperforming LoRA-based unlearning, NPO, and RMU. We therefore use full-weight gradient ascent in the main experiments. We **bold** the lowest binary calibration error.

## C Applying dropout to different layers

We find that the layers which dropout is added to significantly affects the results. In Table 4, we compare the results of adding dropout to the attention layers to adding dropout to the MLP layers that construct the hidden state. We find that adding dropout to attention layers is significantly more effective for models probes trained to recognize uncertainty before the answer and slightly more effective for probes trained to

recognize uncertainty after the answer; this has an intuitive explanation, as the representation of artificial uncertainty is more likely to resemble true uncertainty if noise is added earlier in the process (closer to the input) and then processed normally by the model, rather than vice versa.

| Dataset | Model | Accuracy | Position | Method | Brier↓ | AUROC ↑ | ECE ↓ |
|---------|-------|----------|----------|--------|--------|---------|-------|
| GPQA | Llama-8B | 0.247 | Before | Base | 0.719 | 0.550 | 0.728 |
| | | | | Dropout (attention) | **0.267** | 0.592 | 0.275 |
| | | | | Dropout (hidden) | 0.550 | 0.449 | 0.585 |
| | | | After | Base | 0.494 | 0.685 | 0.551 |
| | | | | Dropout (attention) | **0.227** | 0.624 | 0.179 |
| | | | | Dropout (hidden) | 0.242 | 0.590 | 0.195 |
| MMLU-pro | Llama-8B | 0.366 | Before | Base | 0.552 | 0.558 | 0.565 |
| | | | | Dropout (attention) | **0.252** | 0.563 | 0.125 |
| | | | | Dropout (hidden) | 0.397 | 0.486 | 0.361 |
| | | | After | Base | 0.423 | 0.741 | 0.455 |
| | | | | Dropout (attention) | **0.219** | 0.708 | 0.100 |
| | | | | Dropout (hidden) | 0.233 | 0.657 | 0.091 |

Table 4: Comparing adding dropout to only the attention layers compared to the MLP layers for Llama-8B. We find that adding dropout to attention layers has a more positive effect on our metrics. We **bold** the best-performing Brier score.

## D  Effects of increasing intervention

We find that as the strength of an intervention increases, there is a general trend of initial decrease in Brier score followed by an increase, as can be seen in Figure 3, which plots Brier score of the trained probe[13] compared to the level of dropout for all of our language models. The decrease in Brier score can be attributed to the effects of artificial uncertainty; the following increase can be attributed to divergence of the hidden states. To demonstrate that there is decreasing similarity of representations, we run our model with increasing dropout on GPQA and calculate the similarity of the first hidden state (before the model answers, which would necessarily impact similarity due to differences in the generated answers) with different levels of dropout. We show the results in Table 5, demonstrating that there is increased divergence in the representation as measured by CKA (Kornblith et al., 2019) and average cosine similarity, where higher scores represent increased similarity. We believe this demonstrates the necessity of using methods designed to minimize the divergence of hidden states, as transfer appears to become less effective as divergence increases.

| Dropout rate | CKA | Cosine Similarity |
|--------------|-----|-------------------|
| 0.04 | 0.820 | 0.906 |
| 0.08 | 0.686 | 0.828 |
| 0.12 | 0.618 | 0.755 |
| 0.16 | 0.514 | 0.680 |
| 0.20 | 0.471 | 0.598 |
| 0.24 | 0.379 | 0.493 |

Table 5: Increasing dropout decreases similarity of the intermediate representations used by Llama-8B, which would evidently lead to problems with transfer.

## E  Transfer to open-domain

We examine whether this effect holds true when the trained probes are applied to open-answer questions rather than the original MCQA domain. We test our probes on a 500-question subset of the MATH dataset

---

[13]Trained after the answer

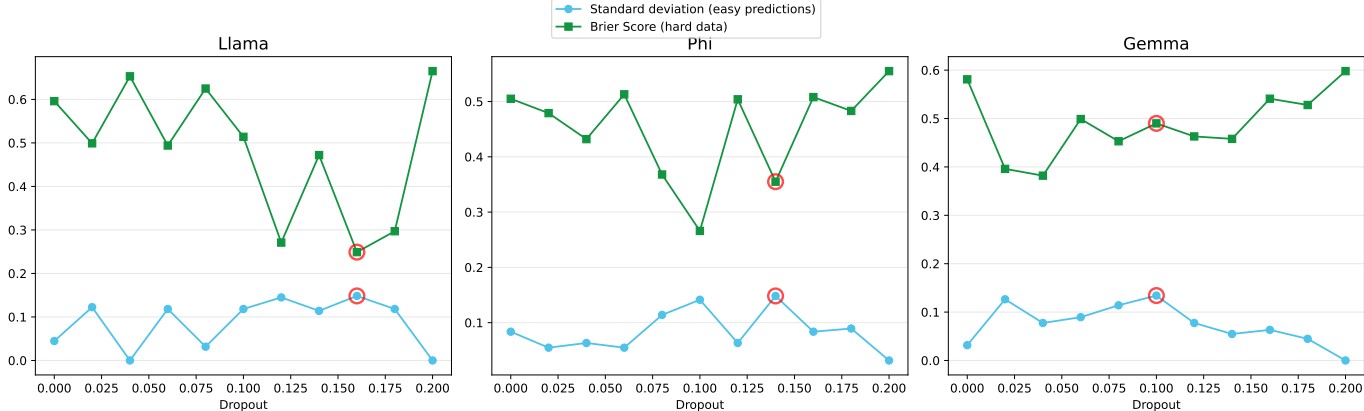

Figure 3: Comparing dropout to Brier score and standard deviation. We find that there is a general trend showing that increasing dropout leads to an improvement in performance followed by increasing Brier score as transfer becomes less effective.

(Hendrycks et al., 2021). To judge correctness, we evaluate the boxed answers using `gemini-2.5-flash` as a judge[14]. We show the results in Table 6, which demonstrates that even without finetuning on this specific domain, the calibration still improves when using artificial uncertainty.

| Dataset | Model | Accuracy | Position | Method | Brier↓ | AUROC ↑ | ECE ↓ |
|---------|-------|----------|----------|--------|--------|---------|-------|
| MATH | LLAMA-8B | 0.330 | AFTER | BASE | 0.499 | 0.767 | 0.523 |
| | | | | UNLEARNING | 0.363 | 0.707 | 0.387 |
| | | | | DROPOUT | **0.237** | 0.687 | 0.184 |

Table 6: Results of transferring the probes trained on MCQA to the MATH dataset. We find that these results are similar to the main-table results, improving calibration on the open domain as well as for multiple choice QA.

## F   Reliability diagrams

We present in Figure 4 the reliability diagrams for our trained probes on MMLU-pro and ARC-Easy.

## G   Epistemic uncertainty

Our main table results have examined the effects of introducing artificial uncertainty through methods which alter the *model's* performance at training time. However, rather than introducing *epistemic* uncertainty through unlearning or dropout, it is possible introduce *aleatoric* uncertainty through the use of ambiguity. Questions which cannot be answered are trivial to construct (e.g. asking a model "Did the coin I just flipped land on heads or tails"), and therefore may serve as a method of introducing uncertainty to train a probe without the need for unlearning or dropout. In this analysis, we train probes on AmbigQA (Min et al., 2020), a Wikipedia QA dataset which annotates ambiguous questions. We train the probe using the ambiguity labels from 1,000 questions from AmbigQA, with the idea being that the model will learn to discriminate between questions the model can and cannot answer. However, the drawback is that this may lead to the probe only recognizing *aleatoric* uncertainty, rather than total uncertainty or epistemic uncertainty.

The results of these experiments can be found in Table 7, where we find that introducing aleatoric uncertainty is an ineffective method of introducing uncertainty. Compared to improving on the base probe in 11/12 settings as with epistemic uncertainty, probes trained on ambiguous data improve in only 8 settings. Notably,

---

[14]String matching may lead to semantically equivalent answers being marked wrong, such as 1/2 vs 0.5

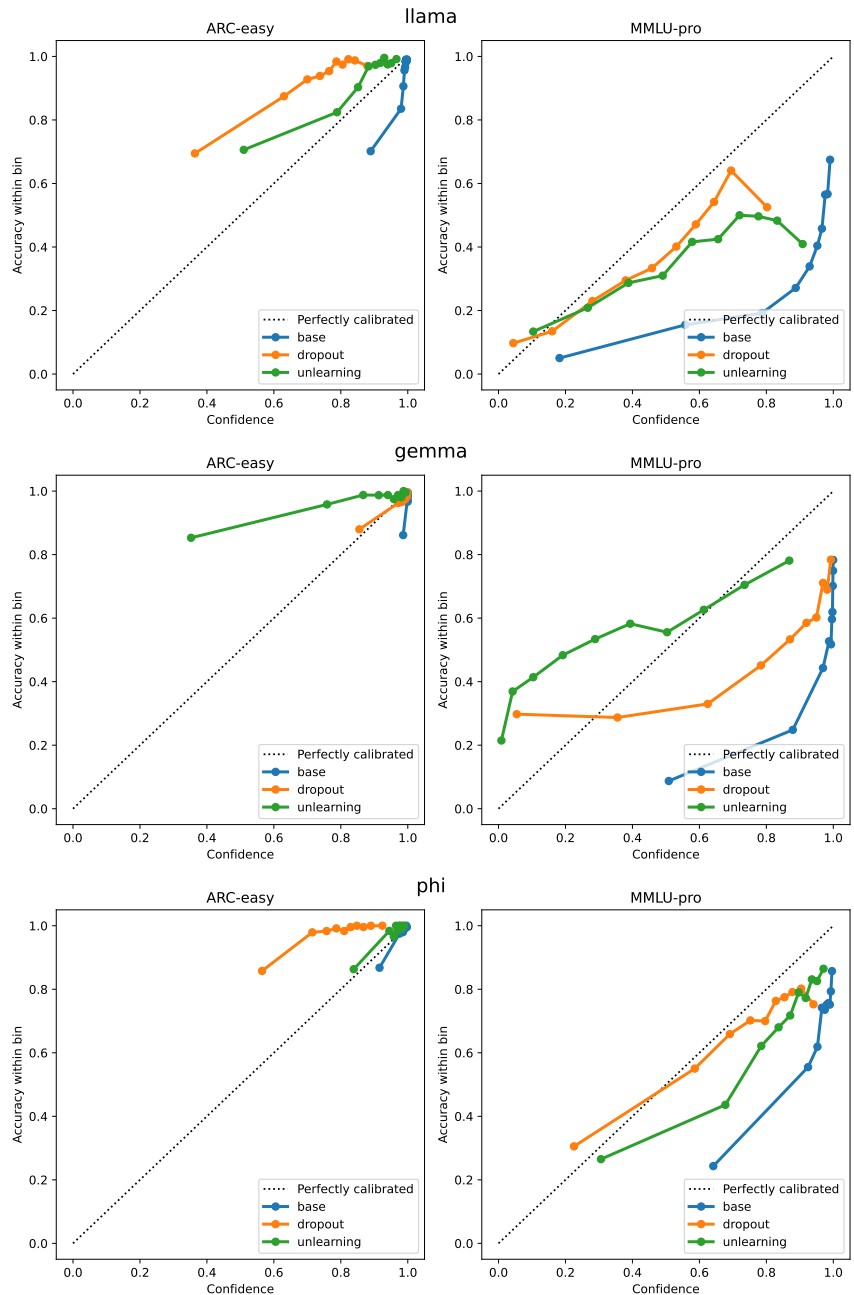

Figure 4: Reliability diagrams (10 bins, quantile binning strategy) on ARC-easy (high accuracy) and MMLU-pro (low accuracy) data. We find that artificial uncertainty noticeably improves calibration on challenging data, and that on easy data decreases calibration comparatively mildly (and in some cases improves upon the base model).

while calibration improves, AUROC often goes down considerably. In 3 settings, AUROC is lowered to the point of being *worse than random chance*, indicating that the changes in Brier score likely represent improved post-hoc calibration with relatively random assignments of scores within that range.

| Dataset | Model | Accuracy | Position | Method | Brier | AUROC | ECE |
|---|---|---|---|---|---|---|---|
| GPQA | Llama-8B | 0.247 | Before | Base | 0.719 | 0.550 | 0.728 |
| | | | | Ambiguous | **0.469** | 0.552 | 0.502 |
| | | | After | Base | 0.494 | 0.685 | 0.551 |
| | | | | Ambiguous | **0.358** | 0.435 | 0.350 |
| | Phi-4 | 0.551 | Before | Base | **0.311** | 0.502 | 0.250 |
| | | | | Ambiguous | 0.428 | 0.517 | 0.429 |
| | | | After | Base | **0.396** | 0.591 | 0.399 |
| | | | | Ambiguous | 0.404 | 0.510 | 0.405 |
| | Gemma-12B | 0.323 | Before | Base | 0.663 | 0.466 | 0.667 |
| | | | | Ambiguous | **0.257** | 0.558 | 0.134 |
| | | | After | Base | 0.598 | 0.486 | 0.610 |
| | | | | Ambiguous | **0.546** | 0.549 | 0.384 |
| MMLU-pro | Llama-8B | 0.366 | Before | Base | 0.552 | 0.558 | 0.565 |
| | | | | Ambiguous | **0.365** | 0.598 | 0.330 |
| | | | After | Base | 0.423 | 0.741 | 0.455 |
| | | | | Ambiguous | **0.332** | 0.559 | 0.287 |
| | Phi-4 | 0.680 | Before | Base | **0.219** | 0.636 | 0.106 |
| | | | | Ambiguous | 0.290 | 0.558 | 0.272 |
| | | | After | Base | **0.265** | 0.683 | 0.261 |
| | | | | Ambiguous | 0.278 | 0.576 | 0.265 |
| | Gemma-12B | 0.526 | Before | Base | 0.453 | 0.467 | 0.448 |
| | | | | Ambiguous | **0.357** | 0.451 | 0.290 |
| | | | After | Base | 0.387 | 0.732 | 0.406 |
| | | | | Ambiguous | **0.348** | 0.684 | 0.356 |

Table 7: Comparing base probe performance to probes trained to recognize ambiguity. We find that this is an unreliable method of improving probe performance, and in particular decreases in AUROC are common and often severe, indicating that discriminating between ambiguous and unambiguous questions and between confident and unconfident answers are too dissimilar to generalize.

# H  Model scaling

Model size can significantly impact calibration (Chhikara, 2025). In this work, we have examined the effects of artificial uncertainty on three LLMs which range from 8 billion parameters to 14 billion parameters. Here, we examine the effects of scaling by looking at Brier score of only predictions on MMLU-pro where *all three models agree*, ensuring that accuracy is not a confounding factor. This results in 780 correct examples and 656 incorrect examples, or a consistent accuracy of 0.543 for each model[15]. We show the Brier score results for each method in Figure 5.

While we examine only three models, these points suggest that increasing scale leads to increased miscalibration on challenging data when accuracy is held consistent. Brier score increases monotonically with model scale, with a large increase with the jump of 4 billion parameters and a smaller increase with another 2 billion parameters. However, we do not note a consistent trend with the probes trained with artificial uncertainty, where performance remains comparatively stable over different model sizes. This suggests that as model size increases, the benefits of artificial uncertainty may be more dramatic.

---

[15]A limitation of this approach is that it likely limits the models to the most extreme examples— those so easy that all models can answer correctly or hard enough that none can answer.

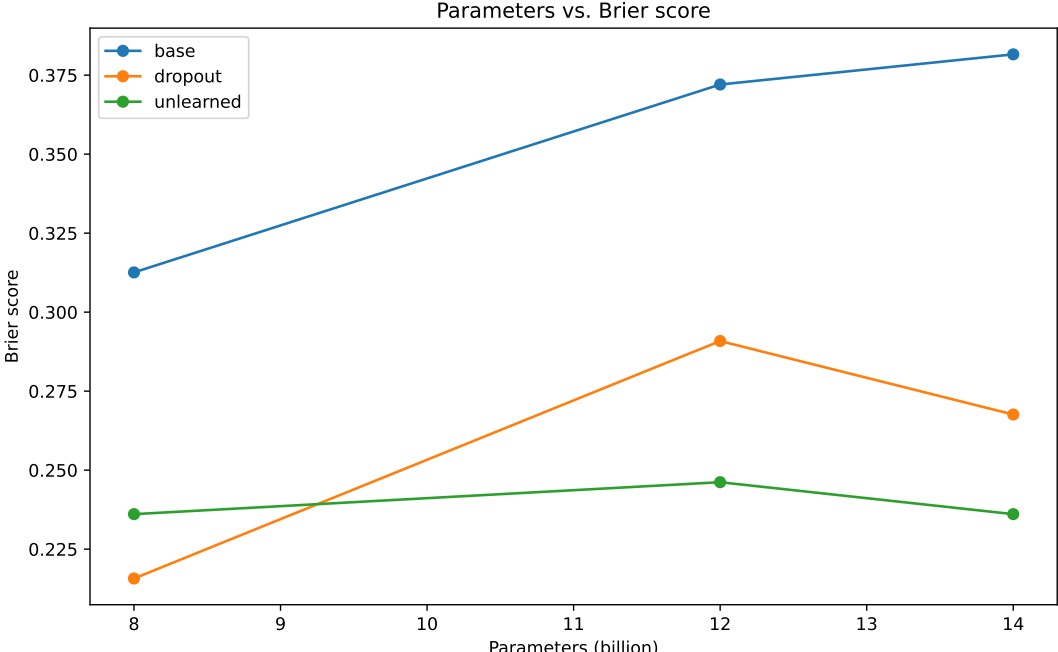

Figure 5: Brier score on filtered data to ensure consistent accuracy plotted against number of parameters. We find that while there is monotonic increases in the Brier score for our base models, Brier score remains comparably consistent for models trained with artificial uncertainty, leading to increasingly large differentials as models increase in scale.

## I Accuracy-controlled performance

In §3.1 we discuss that we are using a weakened version of Assumption 1. Here, we provide support for this decision. We control accuracy strictly on the calibration set, filtering the training data to only the questions that the original model got correctly. We then compare to models trained on a dataset with 90% accuracy, our floor for calibration. We show the results in Table 8, where we find that using 100% accurate training data results in a weak baseline. We therefore compare to a stronger baseline where the model is at least 90% accurate (and thus saturated) but not confined to only 100% accuracy.

## J Prompts

We prompt our model with the following prompt template:

```
Answer the following question.  Enclose concise reasoning in <reasoning> </reasoning>
tags and your FINAL answer in <answer> </answer> tags without any of your work, like
this:  "If each of Lisa's 7 chickens lays 6 eggs, how many eggs does Lisa have?"
A) 24
B) 35
C) 42
D) 50

<reasoning> This can be solved with multiplication.  The answer is 7*6, or 42.
</reasoning> <answer> C) 42 </answer>." Your answer should not include words.
Question:  ...
```

| Model | Dataset | Train Accuracy | Method | Brier↓ | AUROC ↑ | ECE ↓ |
|-------|---------|----------------|--------|--------|---------|-------|
| Llama-8B | GPQA | 90% | Base | 0.542 | 0.636 | 0.582 |
| | | | Unlearned | 0.423 | 0.627 | 0.437 |
| | | | Dropout | **0.395** | 0.703 | 0.422 |
| | | 100% | Base | 0.743 | 0.674 | 0.747 |
| | | | Unlearned | 0.530 | 0.714 | 0.566 |
| | | | Dropout | **0.374** | 0.640 | 0.390 |
| | MMLU-pro | 90% | Base | 0.464 | 0.756 | 0.481 |
| | | | Unlearned | 0.361 | 0.704 | 0.356 |
| | | | Dropout | **0.321** | 0.774 | 0.329 |
| | | 100% | Base | 0.622 | 0.654 | 0.627 |
| | | | Unlearned | 0.459 | 0.743 | 0.468 |
| | | | Dropout | **0.309** | 0.748 | 0.312 |

Table 8: Results comparing the effects of probes trained on data filtered to different accuracies. At 90%, the baseline is notably worse by 20 points for GPQA and 16 points for MMLU-pro. The effects are less prominent for unlearning, and the change in original model's accuracy does not degrade calibration for dropout.

## K Hyperparameters

All probes were trained for 3 epochs on 1000 examples, with batch size of 8 and learning rate of 5 e-4.

| Model | Dropout | Unlearning LR | LoRA rank | LoRA alpha | Unlearning steps |
|-------|---------|---------------|-----------|------------|------------------|
| Llama-8B | 0.16 | 8 E-8 | - | - | 1000 |
| Gemma-12B | 0.10 | 4 E-6 | 8 | 16 | 600 |
| Phi-4 | 0.14 | 7 E-6 | 8 | 16 | 600 |

Table 9: Hyperparameters used in our experiments. Best hyperparameters determined by the method described in §6.1.

