# OpenReview forum: "Inducing Artificial Uncertainty in Language Models"
_TMLR — Under review for TMLR_

### Review · Reviewer_st8n · 2026-06-06

**Summary Of Contributions:**

The authors propose inducing artificial uncertainty, deliberately degrading a copy of the model so that it becomes uncertain even on easy data, training an uncertainty probe on the model's hidden states over easy data, and applying that probe back to the original model on hard data. Then, the author investigates two interventions, gradient-ascent unlearning on the easy set, and dropout in attention layers. And proposes an unsupervised hyperparameter selection criterion: choose the intervention strength that maximizes the variance of the probe's predictions on easy data.

Strengths
1.  The scarcity of uncertain training data is a distinct and worsening problem from the scarcity of hard data. So, the motivation is intersting and clear.

2. The unsupervised hyperparameter selection is the most practically important result. A method that requires hard validation data to tune would defeat the paper's own premise. The variance-maximization criterion closes this loop and is what makes the contribution self-consistent.

3. The easy-data robustness result is valuable. The intuitive prior is that improving hard-data calibration trades off against easy-data calibration. Table 2 showing that probes often match or beat the base model on ARC-Easy is a meaningful and somewhat surprising finding, and the dropout-vs-unlearning tradeoff gives an actionable decision rule.

Weaknesses:

1. Train/test mismatch is acknowledged but not measured. The whole method rests on θ* remaining close enough to θ that a probe transfers. The paper argues this qualitatively but never quantifies representational drift between θ* and θ, nor shows how transfer degrades as the intervention strengthens.

2. The paper motivates itself with GPQA cost and HLE saturation, but tests 8–14B models that are nowhere near saturating the hard sets. Assumption 1 is only weakly instantiated and the authors concede they test a "weakened version" with ~90% easy accuracy, not the ~100% the premise requires. So the evidence supports "this works for moderately-capable models on MCQA," not "this addresses the frontier-saturation problem." Therefore, I think the motivating claim is not directly tested.

3. The loss notation L(ϕ(h_θ), y, ŷ) is introduced but never fully specified.

**Audience:**

Yes

**Audience Explanation:**

This paper targets an important research question: the scarcity of uncertain training data is a distinct and worsening problem from the scarcity of hard data.

**Broader Impact Concerns:**

No concerns.

**Claims And Evidence:**

No

**Claims Explanation:**

See the weakness points 2 and 3.

**Requested Changes:**

1. See the weakness points 1 and 3.

---

### Review · Reviewer_JUu1 · 2026-06-29

**Summary Of Contributions:**

The paper studies uncertainty calibration for LLMs in a setting where only easy or saturated calibration data is available. This is because as models get better, finding or creating challenging datasets where models are uncertain/incorrect becomes harder. To this end, the authors propose inducing "artificial uncertainty" by applying dropout or unlearning to a modified copy of the model, training hidden-state probes on this modified model, and then applying the probes to the original model. Experiments on multiple-choice science QA benchmarks show that the proposed approach generally improves Brier score and ECE compared to probes trained on the original model, although AUROC results are mixed.

**Audience:**

Yes

**Audience Explanation:**

I believe the problem the paper studies is interesting and important and aligns with the interests of some individuals in the TMLR audience. However, as I mentioned above, the technical novelties and experimental scopes may be limited.

**Claims And Evidence:**

Yes

**Claims Explanation:**

The experiments show that the proposed artificial uncertainty methods often improve Brier score and ECE compared to probes trained on the original model, suggesting that dropout and unlearning can improve calibration in the evaluated setting. However, there are several limitations in evaluations.

1. The evidence is less convincing for the stronger claim that the method learns a generally better representation of uncertainty. AUROC results are mixed and often decrease, indicating that the method may mostly rescale or calibrate confidence rather than improve the probe’s ability to distinguish correct from incorrect predictions.

2. In addition, the evaluation scope is quite narrow. The experiments are limited to multiple-choice science QA datasets, where correctness labels are especially clean and answer choices are predefined. The probing approach does not seem inherently restricted to this setting, since it only requires hidden states and binary correctness labels. Broader evaluations on verifiable tasks, such as math reasoning or open-domain QA, would be important for establishing that the method generalizes beyond multiple-choice QA.

3. The paper also lacks sufficient analysis of key factors such as dropout rate, unlearning steps, calibration data size, and calibration-set difficulty. Since the approach depends on inducing an appropriate amount of artificial uncertainty while preserving transferability to the original model, these ablations are central to understanding the method’s effectiveness.

**Requested Changes:**

I suggest the following improvements:

- Clarify the technical contributions and better position the work relative to prior studies on dropout-based uncertainty estimation, hidden-state confidence probes, calibration, and unlearning. In particular, the authors should make clearer which aspects of the proposed approach are novel: the artificial uncertainty framing, the specific probe-training pipeline, or the use of dropout/unlearning for inducing uncertainty.

- Add experiments beyond multiple-choice science QA. The current evaluation is narrow, and the probing setup does not appear inherently restricted to predefined answer choices. Broader evaluations on verifiable tasks such as math reasoning, code generation, or open-domain QA would help establish whether the proposed approach generalizes beyond the current setting.

- Add more analysis and ablations. In particular, the paper should study sensitivity to dropout rate, unlearning steps, calibration data size, and calibration-set difficulty. Since the method depends on inducing an appropriate amount of artificial uncertainty while preserving transferability to the original model, these factors are important for understanding when and why the method works.

---

### Review · Reviewer_JbNY · 2026-07-11

**Summary Of Contributions:**

This work investigates uncertainty in language models, focusing on uncertainty estimation on unseen challenging test data. In particular, this work employs a probing based method to measure the prediction confidence, not directly using the token prediction confidence, for analyzing epistemic uncertainty. Two settings are compared, one for unlearning to increase uncertainty for easy dataset and the other for Monte Carlo dropout. Experiments were conducted using SciQ and CommonsenseQA as easy data, while challenging data was extracted form MMLU and CPQA. The uncertainty introduced by unlearning and dropout is better when measured by Brier score.

Strengths
* This work investigates uncertainty estimation for challenging dataset by artificially introducing uncertainty using only easy dataset. The method is simple: leverage unlearning and dropout for the analysis.
* Experiments show the proposed approach can estimate the uncertainty on challenging dataset without the use of in-domain challenging data for tuning.

Weaknesses
* Several details rely on the cited papers, and the probing method is not clearly described, although it is the main contribution of this study.
* It is not clear whether the claim is supported given the experimental results mainly due to the clarity issue.
* The use of probing is not justified given that the token prediction confidence is the most important for language models and it is merely a proxy.
* This work should justify the use of dropout to a model, which does not employ it during pre-training or post-training. The behavior of dropout during inference for a model trained without dropout is undefined, if my understanding is correct.

**Audience:**

Yes

**Audience Explanation:**

Uncertainty estimation is an important topic in understanding the behavior of language models.

**Claims And Evidence:**

No

**Claims Explanation:**

This work misses several details on the probing and experiment design.

* No detail of the probing method is provided. It is better to explain the method using math notations for clarity.
* Experiments are conducted by unlearning and dropout, but this will have an impact to inference, therefore, the accuracy on test data. However, nothing reported in Table 1. If the accuracy drops due to the injected uncertainty, the experiments are clearly biased.

**Requested Changes:**

* This work needs to describe the details of the probing method in section 3.2, instead of relying on cited papers.
* This work needs further explanation regarding the experiments, in particular, regarding the accuracy.
* This work needs explanation why probing was used for the uncertainty estimation, not the token prediction confidence.
* Justify the use of dropout for a model trained without dropout.